# Improvement of Performances of the Gypsum-Cement Fiber Reinforced Composite (GCFRC)

**DOI:** 10.3390/ma13173847

**Published:** 2020-08-31

**Authors:** Natalia Chernysheva, Valery Lesovik, Roman Fediuk, Nikolai Vatin

**Affiliations:** 1Department of Building Materials, Products and Structures, Belgorod State Technological University Named after V.G.Shoukhov, 46, Kostiukova Str., 308012 Belgorod, Russia; chernysheva56@rambler.ru (N.C.); naukavs@mail.ru (V.L.); 2School of Engineering, Far Eastern Federal University, 8, Sukhanova Str., 690950 Vladivostok, Russia; 3Higher School of Industrial, Civil and Road Construction, Peter the Great St. Petersburg Polytechnic University, 195251 St. Petersburg, Russia; vatin@mail.ru

**Keywords:** gypsum, cement, fly ash, slag, fiber

## Abstract

The novelty of this paper lies in the identification of the scientific patterns of the influence of thermal power plant waste (TPPW) on the hydration mechanism and the structure of the gypsum-cement binder (GCB). The classification of raw materials for the production of GCB has been developed taking into account the genesis, which contributes to the prediction of the properties of composites. The features of the hydration phase formation and hardening of GCB have been studied taking into account the chemical, structural and morphological features of fly ash and slag. In addition, the microstructural, morphological, and thermal properties of the cured binders at a 28 day cure were determined. For the first time, scientific data on the properties of gypsum-cement fiber-reinforced composite using TPPW and microfiber have been obtained. The results show that the synergistic effect of gypsum-cement binder, TPPW, and polyamide or basalt microfiber improves the physicomechanical properties of a 28 day cured binder: compressive strength of 20 MPa, flexural strength of 8.9 MPa, and softening coefficient 0.87.

## 1. Introduction

One of the most important problems of our time is the creation of comfortable conditions for human existence in regard to the use of building structures and materials [1]. At the same time, it is necessary to reduce the energy intensity of the production of composites, expand the range of building materials and effective technologies for their production, taking into account the genesis of raw materials and the sustainability of the “man–material–environment” system [2,3]. Current trends in the development of building materials science are associated with the transition to the creation of multicomponent, multilayer, multilevel composite materials with a given set of properties, their structural and functional organization [4,5]. This ensures their behavior, that they are adapted to variable environmental factors throughout the life of the operation, but also, at least, the preservation or improvement of the quality of the environment [6,7].

Papers in the field of gypsum binders, materials, and products, as well as favorable environmental, technical, and economic aspects of the production and use of these materials indicate that these are all prerequisites for widespread use in construction [8,9,10,11]. In this regard, the development of effective quick-hardening building composites, obtained using new types of available raw materials with improved performance characteristics, is required [12,13]. These requirements are met by water-resistant and frost-resistant gypsum-cement composites, which can reduce the deficit of wall materials, replace energy-intensive cement concrete and shorten the construction time of buildings [14,15,16]. The use of these materials in construction significantly reduces the negative impact on the environment compared to the traditionally used Portland cement (Table 1).

Many research papers have been devoted to solving the problem of managing the structure formation of gypsum and cement materials [20,21,22,23]. At the same time, there are much fewer papers devoted to gypsum-cement binders (GCB) with various additives [24]. In Figure 1, an attempt is made to conduct a generalized analysis of natural and technogenic raw materials that can be used for the production of gypsum-cement binders.

As seen from Figure 1, the conditions for the formation of gypsum binders and mineral additives are the basis for their division into classes. According to this criterion, the materials were divided into natural and man-made. There are three classes of mineral additives, such as crystallization centers, active mineral additives and excipients. The source of the formation of mineral additives are various rocks of sedimentary (diatomite, trepel, opoka), volcanic (tuff, perlite, vermiculite) and metamorphic genesis (quartzite sandstone, ferruginous quartzites), as well as mechanogenic and pyrogenic origin (concrete waste, expanded clay dust, chamotte dust, ash and slag waste).

At the same time, the utilization of industrial waste is a priority for scientists of various fields of knowledge [29,30]. Therefore, the study of the influence of fly ash and hydrodynamic slag from a thermal power plant on the structure formation of a gypsum-cement system is an urgent task.

The second important task that is solved in this article is to increase the strength characteristics of gypsum cement composites using micro-reinforcing fibers. It is known that the use of various fibers, both in cement and in the gypsum matrix, helps to reduce shrinkage cracks by 50–90% [31,32]. Obviously, the fiber content in the gypsum-cement fiber-reinforced composite (GCFRC) will increase its frost resistance and durability characteristics to the level of composites with air-entraining additives.

Thus, the goal of the article is to improve the characteristics of the gypsum-cement fiber reinforced composite using various wastes of the thermal power station and micro-reinforcing fibers.

## 2. Materials and Methods

### 2.1. Characteristics of the Raw Materials Used

Gypsum β-modifications of the grade G-5B II (Belgorod, Russia) was used as a component of the gypsum-cement binder (Table 2).

Portland cement CEM I 42.5 N (Belgorodsky cement, Belgorod, Russia) with a bulk density of 1150 kg/m^3^ was used as a binder material. Fly ash (FA) and slag of the Grozny thermal power plant (TPP) (Grozny, Russia) were used as silica-containing components. The bulk density of fly ash and slag was 900 and 1100 kg/m^3^, respectively. According to the XRD data (Figure 2), the fly ash of the Grozny TPP consists of two modifications of quartz (d = 3.403, 3.247 A), magnetite Fe_3_O_4_ (d = 2.987, 2.600, 2.119, 1.493 A), calcite CaCO_3_ (d = 3.081 A), calcium-iron silicate CaFeSi_2_O_6_. The excess of the background in the region of 20–40° is mainly due to the high iron content.

Mostly (up to 95%) glass phase is present in the slag. The crystalline phase is represented by albite, NaAl-silicates and Al-silicates, several different modifications of quartz, which differ from the natural modifications by the crystal lattice parameters (d = 4.327, 4.106, 3.386, 3.283, 2.477, 2.351, 2.255, 2.140, 1.990, 1.788, 1.681, 1.660, 1. 549, 1.458 A), which is associated with its technogenic origin. The XRD pattern contains peaks of two fairly strong varieties of silicate (albite and anorthoclase K, Na (AlSi_3_O_8_) (d = 3.888, 3.283, 3.23, 2.983 A), except in addition, there are peaks of mica-hydronosodium aluminosilicates (d = 3.527), magnetite Fe_3_O_4_ (d = 3.00, 2.55 A) and calcite CaCO_3_ (d = 3.03, 3.065 A). In the range of 31–32°, the interplanar spacings of sodium aluminum silicates are broadened (AlSi_3_O_8_), and there is also a significant background increase under these interplanar distances (d = 3.383–3.23 A), which indicates the partial presence of their amorphized state (a in Figure 2). As a result of calculations based on the maximum intensity of interplanar distances, of minerals SiO_2_ and K, Na (AlSi_3_O_8_) and the background value, it was found that the content of crystalline SiO_2_ is 56.9%, amorphous SiO_2_—3.7%, amorphous K, Na (AlSi_3_O_8_)—39.4%.

Thus, both in fly ash and in slag, a large number of X-ray amorphous phases are noted. This characterizes the pozzolanic activity necessary for the binding of calcium hydroxide released during the hydration of clinker minerals into secondary hydrosilicates of calcium. The slag particles are layered and consist of dense glassy particles of various sizes of cubic or rounded shape (Figure 3a).

Most of the ash consists of porous glassy particles of various sizes, which have a lamellar, irregular, and angular shape (Figure 3b). Porous, relatively large particles with a concave surface are observed, as well as small particles of a flaky structure. The chemical compositions of the used Portland cement, fly ash, and slag are listed in Table 3.

To improve the rheology, the superplasticizer SP Melment F10 (BASF, Ludwigshafen, Germany) with a bulk density of 550 kg/m^3^ was used. It is optimized to plasticize and reduce water consumption for cement-gypsum mixtures. High modulus basalt and low modulus polyamide fibers were used for micro reinforcing (Table 4). Basalt and polyamide fibers have a density of 2800 and 1100 kg/m^3^, respectively. They are chopped thin fibers with a length of about 10 mm and a diameter of 30–100 µm.

### 2.2. Laboratory Equipment and Research Methods

The ARL X’TRA device (Thermo Fisher Scientific, Waltham, MA, USA) was used for the X-ray diffraction analysis. The microstructure of raw materials and synthesized composites were studied using a Tescan Mira 3 electron microscope (Brno, Czech), which allows not only to obtain SEM images, but also to carry out X-ray fluorescence analysis.

Studies of the composition of hydration products by differential-thermal analysis (DTA) at the age of 28 days were carried out on by the Q600 synchronous thermal analyzer (SDT, Waltham, MA, USA).

The binder components were ground in a LSM ball mill (Moscow, Russia). The specific surface area of the powders was determined on a PSH-2 device (Moscow, Russia). The BET (Brunauer Emmett Teller) area was determined on a Sorbi-M device(Moscow, Russia). Particle size distribution was performed on a MicroSizer 201 device (Moscow, Russia).

The determination of a standard consistency and setting time of the gypsum-cement paste was carried out using the Vik device (Moscow, Russia).

The slump flow of the paste of standard consistency should correspond to 120 ± 5 mm.

To study the physicomechanical characteristics of the obtained gypsum-cement binders, 6 specimens were molded for each composition. The determination of the compressive strength of the GCB was carried out on sample cubes measuring 100 × 100 × 100 mm^3^. Upon reaching the age of 2, 7, and 28 days, as well as after drying to constant weight, the specimens were tested according to EN 12,390-3:2009. Flexural strength of the GCB was determined on beams with a size of 40 × 40 × 160 mm^3^ after 7 and 28 days following EN 1015-11.

The softening coefficient was determined as the ratio of the compressive strength of water-saturated specimens to the compressive strength of dried specimens. In other words, a softening coefficient is actually a coefficient of water resistance, which for gypsum materials is one of the most important indicators of durability.

### 2.3. Mix Design

The preparation of mixes included 3 stages: crushing and drying of a mineral additive (1), mixing of the mineral additive with Portland cement and superplasticizer (2) and the addition of a gypsum binder with short-term grinding of the mix (3). The proportions of the mixtures are gypsum—70%, cement—15%, mineral additive—15%. The amount of superplasticizer 1% by weight of the mix. The amount of water was selected for each composition individually, taking into account the achievement of a standard consistency.

In terms of calcium oxide, the concentration of calcium hydroxide in the liquid phase of the solidifying suspension, regulated by the Russian standard TU 21-31-62-89, on days 5 and 7, respectively, should not exceed 1.1 and 0.85 g/L. For this purpose, two batches of preparations were prepared from 12 samples (6 samples each), with different amounts of mineral additives (Table 5).

The filtered solution was titrated with 0.1 N hydrochloric acid solution in the presence of phenolphthalein. The concentration of CaO (g/L) was determined by the formula:*CaO* = 768 · *A* · *T/B*
where *A* is the amount of hydrochloric acid used for titration, mL; *T* is the titer of hydrochloric acid (HCl content, g/mL); *B* is the amount of the test solution, mL.

As a result of the studies carried out, the concentration of CaO in the solutions was established:-with fly ash (GCB-FA): after 5 days 0.75–1.1 g/L, after 7 days 0.82–0.85 g/L with a cement/additive ratio of 1:1–1:1.5.-with slag (GCB-S): after 5 days 1.04–1.1 g/L, after 7 days 0.64–0.85 g/L with a cement/additive ratio of 1:1–1:1.5.

In the technology of preparing gypsum-cement composites, uniform distribution of fiber is difficult. Therefore, fiber was introduced to the dry mixture of GCB, and then water with the superplasticizer was added, followed by continuous stirring for 5 min until a homogeneous mass was obtained.

### 2.4. Measurement Error

The number of specimens for each tests was 6, while the coefficient of variation of the results did not exceed 5%, and the standard deviation was within 0.95, which ensured the reliability of the results obtained.

## 3. Results and Discussion

### 3.1. Mechanical Activation of the GCB Components

Figure 4 shows the particle sizes distribution of slag and fly ash with a specific surface area of 470 and 690 m^2^/kg, respectively. About 90% of fly ash particles are limited to 18.15–201 µm fractions, and slag particles are limited to 1.1–16.35 µm fractions.

Microporosity depends on the cooling rate of the particles of the feedstock, rapid cooling contributes to this process. Accordingly, in the fly ash compared with the slag revealed a greater number of pores and a large specific surface, as listed in Table 6.

From Table 6, the specific surface of mineral additives determined on the PSH-2 device is not identical to the BET area. This is due to the fact that the microporosity of particles can be not only closed, but also open. The high specific surface of mineral additives indicates their increased reactivity.

Figure 5 shows the particle size distribution of the GCB.

### 3.2. Microstructure of the GCB

Table 7 lists the composition of the products of hydration of the GCB at the age of 28 days, where fly ash is used as a mineral additive.

At the age of two days, a loose structure with a significant number of pores is observed (Figure 6a). At the same time, there are both large pores and small pores between the crystals of new growths. Further, the spaces between the crystals of calcium sulfate dihydrate are filled with ultrafine particles of Portland cement and an active mineral additive, as well as the smallest particles of new growths (Figure 6b). This contributes to an increase in contacts between crystals. After 28 days, under conditions of a solution saturated with lime, calcium hydrosilicates acquire the morphology of dendrite-like formations that create a densified shell around gypsum particles (Figure 6c). When the shell becomes sufficiently dense, the particles are combined into a continuous structure. Due to the hydrosilicates of this morphology, the hardened binder gains strength, its density, water resistance, and durability increase.

Clinker minerals with fly ash reacted more intensely than in the system with slag. A greater amount of Ca(OH)_2_ (d = 4.9 A) and CaCO_3_ (d = 2.088, 1.89 A) are observed. However, these reflections are not visible on XRD patterns, as they overlap with gypsum reflections (Figure 7).

The XRD pattern contains peaks of two strong varieties of silicates (albite and anorthoclase K, Na (AlSi_3_O_8_) (d = 3.888, 3.283, 3.23, 2.983 A).

Figure 8 shows the results of the differential thermal analysis of the developed GCB-FA at the age of 28 days. The endothermic effect at temperatures of 160–220 °C is caused by the dehydration of gypsum CaSO_4_·2H_2_O. The following effect at a temperature of 560 °C is characterized by the decomposition of portlandite. The last endothermic effect at temperatures of 890–910 °C is caused by the dissociation of calcium carbonate CaCO_3_. Two exothermic effects are also observed: at 500 °C, Fe^+2^ is oxidized to Fe^+3^, and at 780–820 °C, CSH (B) decomposes.

Because of this, there is no fundamental difference between the GCB with fly ash and the GCB with slag, further studies of micro reinforcement are given for gypsum-cement binders with fly ash.

### 3.3. Micro Reinforcement

Table 8 lists the results of a study of mortars of a gypsum-cement fiber-reinforced composite (GCFRC).

It was found that the introduction of both polyamide and basalt fibers in the composition of the GCB-FA mix does not affect the setting time, but reduces its slump flow, because some of the mixing water is absorbed by the fiber. Therefore, it is recommended to introduce fibers into the quick-hardening mixture in a moistened state with continuous mixing (Figure 9).

It was found that the introduction of polyamide fiber in an amount of up to 3% is effective. As a result of this, the flexural strength at the age of 7 days is increased by 16%, at 28 days by 12%. Table 9 lists the physicomechanical characteristics of the GCFRC.

With a further increase in the amount of polyamide fiber, CaSO_4_·2H_2_O crystals are formed on their surface, which have insufficient adhesion to polyamide fibers, and no increase in strength was observed.

When basalt fibers are introduced into the GCB-FA, an aggressive medium is formed that destroys their surface with the formation of shells, which increases the adhesion strength of the hardened GCB and basalt fiber and, as a result, the strength of the composite itself. At the age of 7 days, the flexural strength increases by 19%, at the age of 28 days—by 14%, the softening coefficient increases by 17%, with an increase in the durability of the structure. According to the results of the XRD shown in Figure 10, as the hydration products in the samples, there is calcium bicarbonate (d = 7.69; 4.30; 3.81; 3.07; 2.88 A); partially crystallized tobermorite-like calcium hydrosilicate (d = 4.9; 3.07; 2.88; 2.79; 2.42; 1.99; 1.81 A); calcium carbonate (d = 2.5; 2.49; 2.29; 2.09; 1.9 A); portlandite (d = 1.78, 1.67 A); low basic calcium hydroaluminates (d = 3.35, 2.29, 2.22 A).

Thus, micro-reinforcing polyamide and basalt fibers are a kind of substrate on which a strong and dense layer of interfacial transition zone is formed. This layer has a rather large effect on the properties of the obtained quick-hardening material with sufficient saturation of the mix with fibers.

## 4. Conclusions

For the first time, scientific data on the properties of gypsum-cement fiber-reinforced composite using TPPW and microfiber have been obtained. Based on the results of various tests, the following conclusions were drawn:The classification of raw materials for the production of the gypsum-cement binders has been developed taking into account their genesis.The scientific basis for improving the production of water-resistant gypsum composite materials has been supplemented. They consist of the use of semi-aquatic gypsum, Portland cement, and technogenic thermal power plant waste with the grinding of all components.The positive effect of the mechanical activation of the gypsum-cement binder components on the character of particle size distribution has been established. Compared to Portland cement, the GCB has a higher content of fine fractions. Accordingly, when they are used together, small gaps fill the gaps between large ones, compacting, and strengthening the composite structure.The nature of the influence of technogenic silica-containing components on the structure formation of the system “gypsum-cement-superplasticizer-water” has been researched. It consists in the formation of a more dense composite structure due to the synthesis of low-basic calcium hydrosilicates and calcium hydroaluminates in a gypsum matrix. This leads to an increase in the strength, water resistance, and durability of the hardened matrix.It has been established that the addition of polyamide or basalt fibers in the amount of 1–3% into the GCB promotes an increase in flexural strength, water resistance, and a decrease in shrinkage microcracks with an increase in the durability of the structure of the gypsum-cement fiber-reinforced composite.This study is an integral part of large-scale research on the development and use of GCB, conducted by the authors in recent years. The results of previous studies have applications as dry fast-setting mixtures for various construction applications, namely, for the construction of road bases, plumbing cabins, wall products.It would be good to create a future database of results that can be used for more in-depth analysis using artificial intelligence (e.g., artificial neural networks).

## Figures and Tables

**Figure 1 materials-13-03847-f001:**
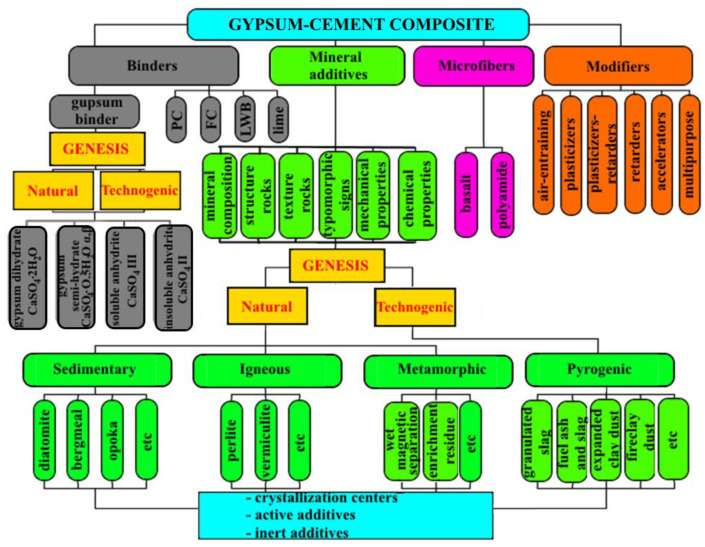
Classification of raw materials for the production of gypsum-cement binders (GCB) [25,26,27,28].

**Figure 2 materials-13-03847-f002:**
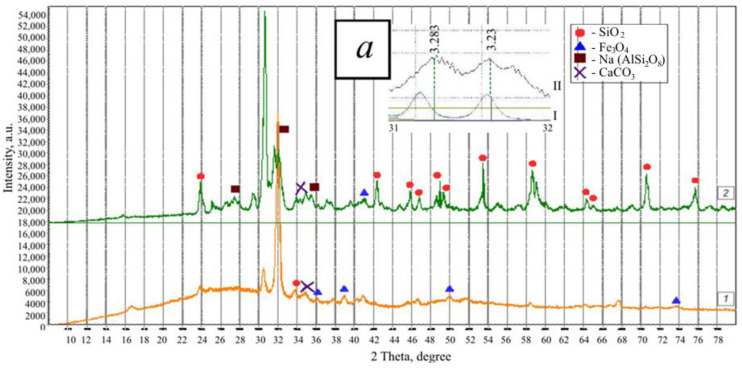
XRD patterns of fly ash (*1*) and slag (*2*) of the Grozny TPP, *a*—comparison of the interplanar distances of Na (AlSi_3_O_8_) in nature (I) and in the slag (II).

**Figure 3 materials-13-03847-f003:**
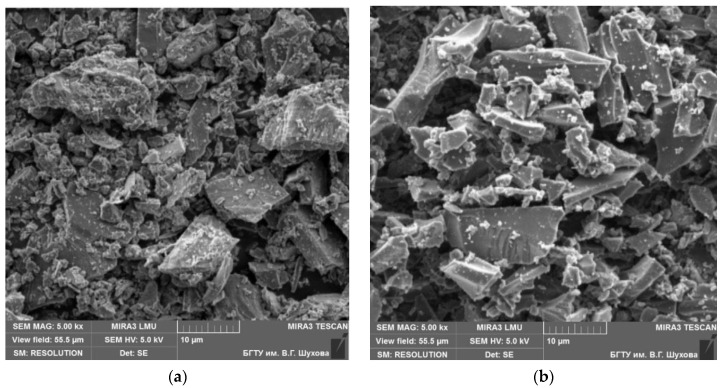
SEM images: (**a**) slag, (**b**) fly ash.

**Figure 4 materials-13-03847-f004:**
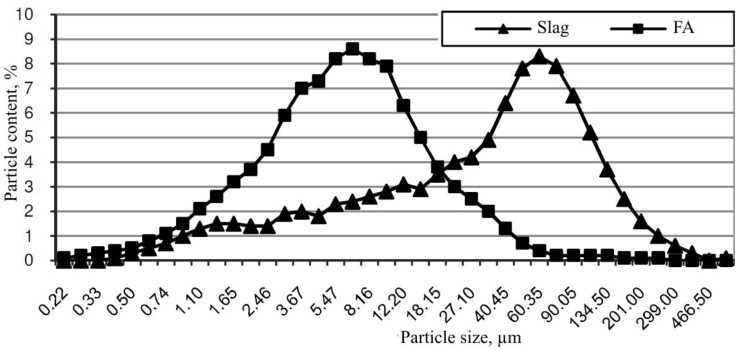
Fly ash (FA) and slag Particle size distribution.

**Figure 5 materials-13-03847-f005:**
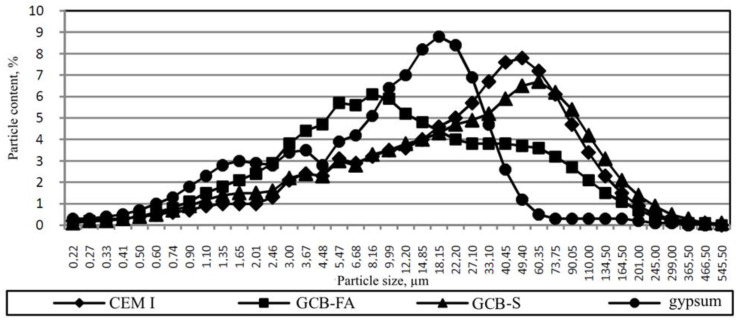
Particle size distribution of the GCB.

**Figure 6 materials-13-03847-f006:**
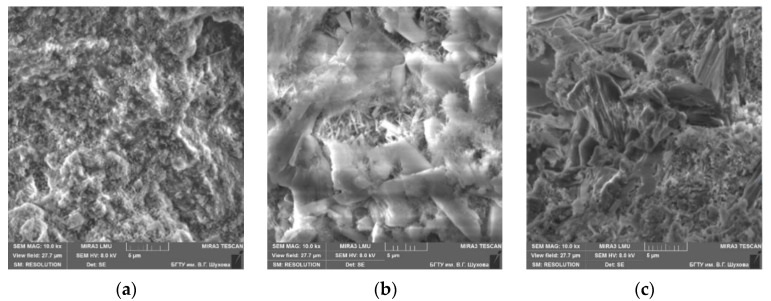
The microstructure of the hardened GCB-FA. at the age of 2 (**a**), 7 (**b**) and 28 (**c**) days.

**Figure 7 materials-13-03847-f007:**
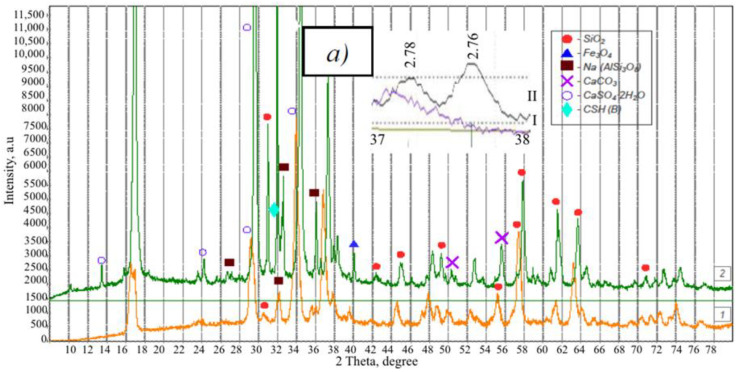
XRD patterns of the GCB-FA (*1*) and GCB-S (*2*); *a*—comparison of the profiles of the main interplanar distances of alite and belite in the GCB-FA (I) and GCB-S (II).

**Figure 8 materials-13-03847-f008:**
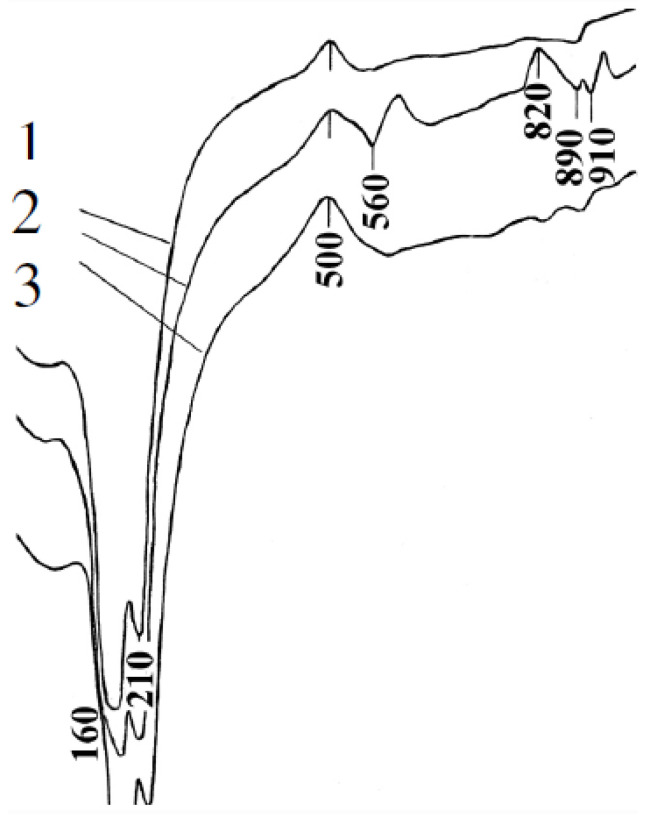
Results of the differential thermal analysis of the developed GCB at the age of 28 days. 1—GCB; 2—GCB-FA; 3—GCB-S. Three-digit numbers represent degrees Celsius.

**Figure 9 materials-13-03847-f009:**
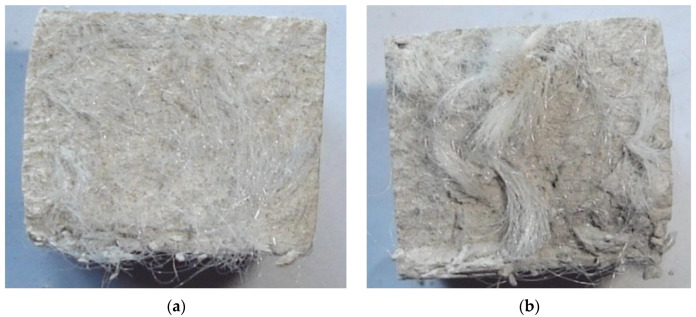
Uniform (**a**) and uneven (**b**) distribution of polyamide fiber in hardened GCB-FA specimens.

**Figure 10 materials-13-03847-f010:**
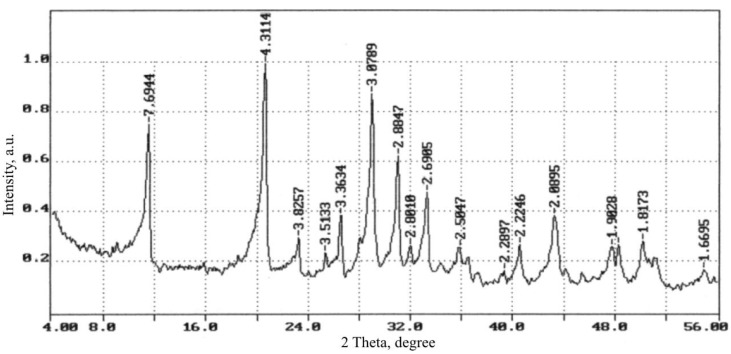
XRD pattern of the hardened GCB-FA. Identical for both types of fiber.

**Table 1 materials-13-03847-t001:** The environmental aspect of the production of mineral binders [17,18,19].

Type of Binder	Annual Production in the World, Million Tons	Energy Intensity		Emissions			
		Production of 1 t, kg Fuel Equivalent	Total, Mill. t Fuel Equivalent	CO_2_		Dust	
				g/t	Total, Mill. t	g/t	Total, Mill. t
Cement	3700	150	555	50	2159	21	53.34
Lime	340	204	69.36	28	151	5	1.43
Gypsum	152	47	7.144	-	-	-	-

**Table 2 materials-13-03847-t002:** Characteristics of the gypsum used.

Bulk density, kg/m^3^	650
Fineness as residue on the sieve of 0.2 mm, %	10
Standard consistency	0.48
Setting time, min-sec	
start	3-00
end	12-00
Flexural strength, MPa	2.4
Compressive strength, MPa	6.0
dry to constant weight	11.6
water saturated	5.3
Softening coefficient	0.45

**Table 3 materials-13-03847-t003:** Technical characteristics of the micro-reinforcing fibers.

Chemical Composition	CaO	SiO_2_	Al_2_O_3_	Fe_2_O_3_	MgO	SO_3_	Alkalis
CEM I, %	65.5	21.8	4.9	4.0	1.2	-	0.6
Fly ash, %	31.4	32.4	5.6	31.4	2.3	1.3	2.9
Slag, %	2.6	67.5	15.0	2.6	0.5	0.2	7.8

**Table 4 materials-13-03847-t004:** Technical characteristics of the micro-reinforcing fibers.

Fiber	Density, kg/m^3^	Tensile Strength, MPa	Elastic Modulus, MPa	Elongation at Break, %
Polyamide	900	720–750	1900–2000	24–25
Basalt	2600–2700	1600	8000–11,000	1.4–3.6

**Table 5 materials-13-03847-t005:** Compositions of preparations for determining the effect of mineral additives on the concentration of CaO in aqueous gypsum-cement suspensions.

Sample	Amount of Materials for Creating of Preparations, g			
	Gypsum	Portland Cement	Mineral Additive	Distilled Water
1	4	2.5	0.625	100
2	4	2.5	1.25	100
3	4	2.5	2.5	100
4	4	2.5	3.75	100
5	4	2.5	5	100
6	4	2.5	6.25	100

**Table 6 materials-13-03847-t006:** Properties of finely ground fly ash and slag.

Properties	Fly Ash	Slag
Specific surface area determined by a PSH-2 device, m^2^/kg	690	470
BET area determined by a Sorbi-M device, m^2^/g	72.00	1.64
Pore volume with radius less than 19.4 nm, cm^2^/g	0.018	0.003

**Table 7 materials-13-03847-t007:** The composition of the hydration products of the gypsum-cement binder- fly ash (GCB-FA) at microprobe points.

Elements	Content of Elements, wt.% at Microprobe Points				
	1	2	3	4	5
C	79.8	30.8	47.0	16.2	83.3
O	9.8	42.8	32.8	44.1	13.9
Si	1.4	3.8	3.2	5.5	0.7
Ca	6.7	20.2	12.2	22.2	2.2
Fe	0.6	0.2	2.3	5.2	-
Mg	1.1	0.1	-	1.7	-
S	0.7	0.8	0.7	2.1	-
Na	-	0.1	-	-	-
K	0.1	0.3	0.5	0.5	-
Al	0.7	0.7	1.5	4.0	-

**Table 8 materials-13-03847-t008:** Effect of microfiber on the properties of the hardened GCB-FA.

Fibers, %		Slump Flow, mm	Setting Time, min-s	
Polyamid	Basalt		Start	End
-	-	140	5-00	8-30
1	-	100	5-00	8-30
3	-	90	5-00	8-30
5	-	75	5-00	8-30
-	1	120	5-00	8-30
-	3	110	5-00	8-30

**Table 9 materials-13-03847-t009:** Physicomechanical characteristics of the gypsum-cement fiber-reinforced composite (GCFRC).

Fibers, %		Density, kg/m^3^	Flexural Strength, MPa		Compressive Strength, MPa				Softening Coefficient, %
Polyamid	Basalt		7 Days	28 Days	2 Days	7 Days	28 Days	Dried	
-	-	1170	6.2	7.8	4.9	15.6	18.0	20.0	0.74
1	-	1270	6.6	8.3	5.3	17.3	19.1	20.4	0.81
3	-	1370	7.2	8.7	5.4	19.2	19.8	23.8	0.83
5	-	1550	6.7	8.3	5.0	15.7	18.3	20.2	0.83
-	1	1340	6.8	8.6	5.6	17.8	19.5	21.5	0.84
-	3	1410	7.4	8.9	6.7	19.8	20.0	24.0	0.87

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
