# Peer review of "Improvement of Performances of the Gypsum-Cement Fiber Reinforced Composite (GCFRC)"

_materials, 2020, doi:10.3390/ma13173847_

Round 1

Reviewer 1 Report

The paper reports and discusses the results of an experimental analysis carried out with the aim of identifying the influence of thermal power plant waste (TPPW) on the hydration mechanism and the structure of the  gypsum-cement binder (GCB). The experimentation carried out shows interdisciplinary characteristics and the results obtained are innovative. The topic is interesting because it deals with a binder, gypsum, little studied and little used, but which has a lot of potential.

In the opinion of this reviewer, the scope of the study is aligned with the journal as well and the paper is written in proper English. Then I recommend it for publication after a minor revision.

Some suggestions are given:

  • The following sentence is reported in the abstract: “… and polyamide or basalt microfiber improves the physicomechanical properties of a 28-day cured binder: compressive  strength of 20 MPa, flexural strength of 8.9 MPa, and softening coefficient 0.87”. In table 8, the values reported in the sentence appear to be relative to fiber-free samples. It's correct? Or maybe table 8 is wrong.

  • With regard to the compression tests carried out to determine the compressive strength, it would be very interesting to also report the stress-strain diagrams as well as an explanation of the test apparatus and methods of carrying out it. The comparison between the stress-strain diagrams of the different samples allows to draw important considerations on the mechanical behavior of the different materials.

  • In fact, even if the authors do not want to calculate, in addition to the compressive strength, other mechanical parameters such as the stiffness of the initial branch and the strain capacity before and after the peak (if the test is carried out with imposed displacement), the comparative observation of the diagrams allows to draw qualitative considerations on these further parameters.

  • As regards the data obtained from the mechanical tests, from table 8 it is possible to understand that for some mixes, the number of samples (those with polyamid and basalt fibers) subjected to test is equal to 1. While only three is the number of those not additives with fibers. A sentence should be inserted that specifies that these results are preliminary and indicative because they have no statistical validity (especially those made on a single sample). If, on the other hand, I have misunderstood and a greater number of samples of the same type were carried out, it is necessary to specify in the table: number of samples, mean value, standard deviation and coefficient of variation).

  • Still in relation to table 8, the mechanical values obtained for the samples with the addition of fibers are internal to the variation of the values obtained without fibers and seem very related to the Density value (obviously). A comment on this should be added

  • In the introduction, with reference to the following sentence “Papers in the field of gypsum binders, materials, and products, as well as favorable 40 environmental, technical, and economic aspects of the production and use of these materials 41 indicate that there are all prerequisites for widespread use in construction[8–10]”, it might be useful to introduce the reference below, in which gypsum is used as a matrix for a fiber-reinforced composite used as a reinforcement for masonry:
    Rovero, L., Galassi, S., & Misseri, G. (2020). Experimental and analytical investigation of bond behavior in glass fiber-reinforced composites based on gypsum and cement matrices. Composites Part B: Engineering, Volume 194, 1 August 2020, 108051

Author Response

Dear Reviewer!

Thank you for your interest in my manuscript. Thanks to your valuable comments, the manuscript just got better! All comments were carefully analyzed and used for revising the manuscript. All changes to the manuscript were highlighted in green.

Responses to Reviewer comments:

Comment 1: The following sentence is reported in the abstract: “… and polyamide or basalt microfiber improves the physicomechanical properties of a 28-day cured binder: compressive strength of 20 MPa, flexural strength of 8.9 MPa, and softening coefficient 0.87”. In table 8, the values reported in the sentence appear to be relative to fiber-free samples. It's correct? Or maybe table 8 is wrong.

Response 1: Yes, the percentage of two types of fiber in this table has been adjusted. According to the new numbering, this is Table 9.

Comment 2: With regard to the compression tests carried out to determine the compressive strength, it would be very interesting to also report the stress-strain diagrams as well as an explanation of the test apparatus and methods of carrying out it. The comparison between the stress-strain diagrams of the different samples allows to draw important considerations on the mechanical behavior of the different materials.

Response 2: Unfortunately, the length of the article does not allow the introduction of additional materials. However, the compressive strength results are given in sufficient detail.

Comment 3: In fact, even if the authors do not want to calculate, in addition to the compressive strength, other mechanical parameters such as the stiffness of the initial branch and the strain capacity before and after the peak (if the test is carried out with imposed displacement), the comparative observation of the diagrams allows to draw qualitative considerations on these further parameters.

Response 3: The research you proposed is quite voluminous and will form the basis for continuing research in the next manuscript.

Comment 4: As regards the data obtained from the mechanical tests, from table 8 it is possible to understand that for some mixes, the number of samples (those with polyamid and basalt fibers) subjected to test is equal to 1. While only three is the number of those not additives with fibers. A sentence should be inserted that specifies that these results are preliminary and indicative because they have no statistical validity (especially those made on a single sample). If, on the other hand, I have misunderstood and a greater number of samples of the same type were carried out, it is necessary to specify in the table: number of samples, mean value, standard deviation and coefficient of variation).

Response 4: The number of specimens for each tests was 6, while the coefficient of variation of the results did not exceed 5%, and the standard deviation was within 0.95, which ensured the reliability of the results obtained. (section 2.4)

Comment 5: Still in relation to table 8, the mechanical values obtained for the samples with the addition of fibers are internal to the variation of the values obtained without fibers and seem very related to the Density value (obviously). A comment on this should be added

Response 5: The density of the specimens changes due to the addition of fiber. However, it is not entirely correct to say that it is the density that affects the mechanical characteristics. It is micro-reinforcement of the composite that has the greatest influence.

Comment 6: In the introduction, with reference to the following sentence “Papers in the field of gypsum binders, materials, and products, as well as favorable 40 environmental, technical, and economic aspects of the production and use of these materials 41 indicate that there are all prerequisites for widespread use in construction[8–10]”, it might be useful to introduce the reference below, in which gypsum is used as a matrix for a fiber-reinforced composite used as a reinforcement for masonry:

Rovero, L., Galassi, S., & Misseri, G. (2020). Experimental and analytical investigation of bond behavior in glass fiber-reinforced composites based on gypsum and cement matrices. Composites Part B: Engineering, Volume 194, 1 August 2020, 108051

Response 6: This reference has been added in the place you specified as [11]

Reviewer 2 Report

The paper presents an experimental study of gypsum-cement-binder (GCB) reinforced with fly ash or microfibers of polyamide or basalt. Chemical, structural, microstructural and morphological features are detailed, and the SEM and XRD data provide detailed analysis of these features. The paper is technically sound and the results will be of interest to the field. I have several editorial comments the authors might address:
1. The paper is well presented until Section 3.3 –results are shown for the GCFRC but no details are provided in Materials regarding the fibre properties, nor in the Methods regarding how the specimens were prepared. It would be useful to add such details.
2. In the conclusions the “water-resistance” and “durability” are referred to several times. Presumably these characteristics are inferred from the softening coefficient – it would be useful to add a comment on how these characteristics were assessed.
3. CGB should be GCB in Fig 5
4. it would read better if there was consistency in terminology in the tables and figures captions – eg Figs 5 to 7 and Table 6
5. Line 246 suggests the microfibre study was conducted on GCB with fly ash, but this is not referred to thereafter – was it plain GCB or GCB-FA?
6. Line 257 – please clarify the statement “part of the mixing water of the fiber is taken over.”
7. Table 8 – it appears there is an error in the magnitudes of Fibers % – currently it says both polyamide and basalt fibres were added at the same time?
8. Line 276 – “tensile strength” should be “flexural strength”?
9. Line 278 – Fig 9 should be Fig 10?

Author Response

Dear Reviewer!

Thank you for your interest in my manuscript. Thanks to your valuable comments, the manuscript just got better! All comments were carefully analyzed and used for revising the manuscript. All changes to the manuscript were highlighted in green.

Responses to Reviewer comments:

Comment 1: The paper is well presented until Section 3.3 –results are shown for the GCFRC but no details are provided in Materials regarding the fibre properties, nor in the Methods regarding how the specimens were prepared. It would be useful to add such details.

Response 1: The characteristics of the fibers used have been added to Table 4. And the procedure of mixing composites with the introduction of fiber was added on the lines 187-190.

Comment 2: In the conclusions the “water-resistance” and “durability” are referred to several times. Presumably these characteristics are inferred from the softening coefficient – it would be useful to add a comment on how these characteristics were assessed.

Response 2: The relationship between softening coefficient, water resistance and durability is shown on lines 157-159.

Comment 3: CGB should be GCB in Fig 5.

Response 3:It has been changed

Comment 4: it would read better if there was consistency in terminology in the tables and figures captions – eg Figs 5 to 7 and Table 6.

Response 4: Consistent labeling of composites has been implemented in all specified locations

Comment 5: Line 246 suggests the microfibre study was conducted on GCB with fly ash, but this is not referred to thereafter – was it plain GCB or GCB-FA?

Response 5: Yes, that's right, in this section all the compositions with ash. Corresponding adjustments were made to lines 268, 271, 276, 289, 300.

Comment 6: Line 257 – please clarify the statement “part of the mixing water of the fiber is taken over.”

Response 6: It has been changed to «some of the mixing water is absorbed by the fiber» (lines 271-272)

Comment 7: Table 8 – it appears there is an error in the magnitudes of Fibers % – currently it says both polyamide and basalt fibres were added at the same time?

Response 7: The error has been revised (See Table 9)

Comment 8: Line 276 – “tensile strength” should be “flexural strength”?

Response 8: It has been changed (line 292)

Comment 9: Line 278 – Fig 9 should be Fig 10?

Response 9: It has been changed (line 294)

Reviewer 3 Report

The authors present a work on the Improvement of performances of the gypsum-cement fiber reinforced composite (GCFRC). The subject of the authors work is an important significant issue in structural engineering and materials. 

The literature review is written correctly in terms of content. It contains interesting information referring to the topic of the article. The list of literature contains 31 items, but a large part of them does not refer to gypsum composites but to concrete. In many cases, the contained literature items are new. It is important because it increases the substantive value of this part of the article.

The scope and description of the research presented below is comprehensible and factually correct. However, I think that the authors should attach a few photographs of samples used for testing, testing equipment and samples during or after testing, e.g. strength.

The analysis of the results and their discussion is carried out correctly in terms of content. I have no comments. I would like to add that the authors have created a very interesting database of results. The results presented in the article are a solid and good starting point for future in-depth research in this very promising field of research, very important for construction practice. I think it would be good to create a future database of results that can be used for more in-depth analysis using artificial intelligence (e.g. artificial neural networks).

In my opinion, there should be a reference to construction practice in the summary. Were the authors' results used in practice, to make some elements on the construction site or to make building materials?

In my opinion, the article requires minor additions. Before publication in Materials journal, please make the following corrections:

- please add photos of the samples tested,

- please add some of the test equipment together with the samples during testing,

- please show examples of samples after strength testing,

- please write in summary what can be the practical use of the results obtained in the construction practice?

In summary, I conclude that the article is important from the point of view of construction practice.

After these few additions, the article can be published.

Author Response

Dear Reviewer!

Thank you for your interest in my manuscript. Thanks to your valuable comments, the manuscript just got better! All comments were carefully analyzed and used for revising the manuscript. All changes to the manuscript were highlighted in green.

Responses to Reviewer comments:

Comment 1: The scope and description of the research presented below is comprehensible and factually correct. However, I think that the authors should attach a few photographs of samples used for testing, testing equipment and samples during or after testing, e.g. strength.

Response 1: Photographs of specimens after bending tests are shown in Fig. 9.

Comment 2: The analysis of the results and their discussion is carried out correctly in terms of content. I have no comments. I would like to add that the authors have created a very interesting database of results. The results presented in the article are a solid and good starting point for future in-depth research in this very promising field of research, very important for construction practice. I think it would be good to create a future database of results that can be used for more in-depth analysis using artificial intelligence (e.g. artificial neural networks).

Response 2: Thank you for the appreciation of our work. Your wishes for identifying prospects for further research have been added to point 7 of the conclusions.

Comment 3: In my opinion, there should be a reference to construction practice in the summary. Were the authors' results used in practice, to make some elements on the construction site or to make building materials?

Response 3: This information has been added to point 6 of conclusions.

Comment 4: In my opinion, the article requires minor additions. Before publication in Materials journal, please make the following corrections:

- please add photos of the samples tested,

- please add some of the test equipment together with the samples during testing,

- please show examples of samples after strength testing,

- please write in summary what can be the practical use of the results obtained in the construction practice?

Response 4: All these comments were carefully considered and addressed. See responses to comments 1-3.